# Hydrogen/Deuterium Exchange Mass Spectrometry for Probing the Isomeric Forms of Oleocanthal and Oleacin in Extra Virgin Olive Oils

**DOI:** 10.3390/molecules28052066

**Published:** 2023-02-22

**Authors:** Ramona Abbattista, Ilario Losito, Graziana Basile, Andrea Castellaneta, Giovanni Ventura, Cosima Damiana Calvano, Tommaso R. I. Cataldi

**Affiliations:** 1Dipartimento di Chimica, Università degli Studi di Bari “Aldo Moro”, Via E. Orabona 4, 70126 Bari, Italy; 2Centro Interdipartimentale SMART, Università degli Studi di Bari “Aldo Moro”, Via E. Orabona 4, 70126 Bari, Italy

**Keywords:** secoiridoids, oleocanthal, oleacin, olive oil, high resolution mass spectrometry, H/D exchange

## Abstract

Reversed-phase liquid chromatography and electrospray ionization with Fourier-transform single and tandem mass spectrometry (RPLC-ESI-FTMS and FTMS/MS) were employed for the structural characterization of oleocanthal (OLEO) and oleacin (OLEA), two of the most important bioactive secoiridoids occurring in extra virgin olive oils (EVOOs). The existence of several isoforms of OLEO and OLEA was inferred from the chromatographic separation, accompanied, in the case of OLEA, by minor peaks due to oxidized OLEO recognized as oleocanthalic acid isoforms. The detailed analysis of the product ion tandem MS spectra of deprotonated molecules ([M-H]^−^) was unable to clarify the correlation between chromatographic peaks and specific OLEO/OLEA isoforms, including two types of predominant dialdehydic compounds, named *Open Forms II*, containing a double bond between carbon atoms C8 and C10, and a group of diasteroisomeric closed-structure (i.e., cyclic) isoforms, named *Closed Forms I*. This issue was addressed by H/D exchange (HDX) experiments on labile H atoms of OLEO and OLEA isoforms, performed using deuterated water as a co-solvent in the mobile phase. HDX unveiled the presence of stable di-enolic tautomers, in turn providing key evidence for the occurrence, as prevailing isoforms, of *Open Forms II* of OLEO and OLEA, different from those usually considered so far as the main isoforms of both secoiridoids (having a C=C bond between C8 and C9). It is expected that the new structural details inferred for the prevailing isoforms of OLEO and OLEA will help in understanding the remarkable bioactivity exhibited by the two compounds.

## 1. Introduction

Epidemiological studies in the last two decades have pointed out that a high dietary intake of extra virgin olive oil (EVOO) is associated with lower incidences of cancer, cardiovascular, metabolic, and Alzheimer’s diseases [1,2,3,4,5]. Notably, EVOO nutraceutical properties are not shared by other vegetable oils since they are mainly related to olive-specific phenolic compounds, in particular *secoiridoids*, which are secondary metabolites unique to *Oleaceae* plants and arise from the monoterpene pathway [6]. Specifically, the drupes and leaves of *Olea europaea* are rich in oleuropein and ligstroside, whose chemical structure includes an ester linkage between the glucosidic form of the iridoid known as elenolic acid and the phenolic compounds hydroxytyrosol and tyrosol, respectively (see the upper left corner of Figure 1). As an example of the peculiar bioactivity exhibited by olive secoiridoids, oleuropein has been claimed to mitigate cisplatin-induced acute injury in the kidney and liver of mice through the formation of a conjugate with a metabolite of the drug, contributing to reducing its toxicity [7]. 

In addition, upon olive drupe crushing during EVOO production, the detachment of glucose, catalyzed by endogenous β-glucosidase, and further chemical processes lead to a complex set of isoforms of the aglycones of oleuropein (OA) and ligstroside (LA) (see Figure 1). OA and LA major isoforms have been recently investigated in detail in our laboratory using liquid chromatography coupled with mass spectrometry [8,9]. Notably, two further secoiridoids, oleac(e)in (OLEA) and oleocanthal (OLEO), can be generated during EVOO production from OA and LA, respectively, through a reaction catalyzed by further endogenous enzymes of the olive plant, namely methylesterases, followed by decarboxylation [10]. Starting from this background, major isoforms expected for OLEA and OLEO, directly descending from those recently unveiled for OA and LA [8,9], have been reported in Figure 1. Identified for the first time in EVOO by Montedoro et al. in 1993 [11], OLEA and OLEO subsequently gained general interest for their anti-inflammatory activity, as evidenced by Beauchamp et al. in 2005 [12]. Since then, other important bioactivities, such as anti-cancer [13] and anti-Alzheimer effects [14,15], as well as a protective role against arthropathy [16] and cardiovascular diseases [17], have been discovered, especially for OLEO. The identification of the ibuprofen-like anti-inflammatory property, exerted by the significant reduction of cyclooxygenase (COX2) activity and of the inflammation cascade in humans, has been the most intriguing bioactivity attested for OLEA and OLEO [18]. Notably, OLEO also elicits an unusual oral pungency, sensed almost exclusively in the throat [19,20], and protects EVOO from rancidity during storage thanks to its strong antioxidant activity [21]. Interestingly, secoiridoids such as those found in EVOO are also involved in the olive plant defense system, acting as phytoalexins, which are antimicrobial/antifungal low-molecular-weight compounds synthesized de novo following a pathogenic attack [22]. According to the literature, the most relevant role in determining such bioactivity, as well as the anti-cancer and anti-inflammatory effects exerted by EVOO secoiridoids, can be ascribed to the glutaraldehyde-like structure assumed by stable isoforms of EVOO secoiridoids bearing two aldehydic groups on their molecular structure, such as those classified as *Open Forms I* and *II* in Figure 1. Indeed, the dialdehydic form of OLEA has displayed significant anti-breast cancer properties [23] and potent antioxidant activity, even better than those of hydroxytyrosol [24]. The aldehydic moieties of OLEA and OLEO are also able to interact with proteins as a part of the plant defense process against herbivores and pathogens [22]. They have also been found to be sensitive to oxidation during EVOO storage, being turned into carboxylic groups [25]. 

In recent years, the structure-related bioactivity that emerged in OLEA and OLEO has urged the characterization of their isoforms. The two compounds have been generally represented as the esters formed by an open-structure isoform of decarboxymethylated elenolic acid (EDA) and tyrosol (p-hydroxy-phenylethyl alcohol, p-HPEA) or 3-hydroxytyrosol (3,4-dihydroxy-phenylethyl alcohol, 3,4-DHPEA), respectively [26,27], i.e., as the isoforms classified as *Open Forms I* in Figure 1. In fact, early spectroscopic data [11] led scientists to hypothesize the existence of a dialdehydic open structure with a C=C bond located between C8 and C9 carbon atoms, according to the numbering adopted in Figure 1. The same structure was proposed by Karkoula et al., whose NMR studies, coupled with reactivity tests performed in various organic solvents, pointed out the presence of aldehydic groups in the OLEA and OLEO structures, due to the formation of closely related hemiacetalic and/or acetalic derivatives in solution [28]. Attempts to separate the two isomers expected for each of the two secoiridoids, related to the geometry of the C=C bond between the C8 and C9 atoms, were carried out using liquid chromatography coupled with UV, NMR, or MS. However, only broad chromatographic peaks or peaks with shoulders were obtained [29,30]. The detection of two chromatographically resolved isomers of OLEO was shown for the first time by Impellizzeri and Lin in 2006 [31], suggesting the existence of *cis*/*trans* (E/Z) geometrical isomers due to the C8=C9 double bond. This hypothesis was confirmed by Adhami et al. [32], who were able to separate OLEO isomers as two closely eluting peaks and characterize them distinctly using HPLC with UV and NMR detection. Interestingly, a subsequent LC-MS study on EVOOs produced from Spanish olive varieties [33] revealed that at least two chromatographic peaks for both OLEO and OLEA, although not resolved, were observed when an acetonitrile-based, instead of a methanol-based, elution gradient was adopted for RPLC separation.

The scenario described so far changed recently after more than two chromatographic peaks were detected for both OLEO and OLEA in EVOO extracts [25] using the RPLC-ESI-FTMS and MS/MS methods previously adopted to characterize oleuropein and ligstroside aglycones in EVOO [8,9], based on acetonitrile as the organic solvent in the mobile phase. Due to the inherent complexity of OA and LA isoforms, whose MS/MS patterns were found to be generally similar, the method had to be integrated through hydrogen/deuterium exchange (HDX) experiments. The latter proved to be a powerful tool, leading to the unveiling of the occurrence of stable enolic tautomers for some OA and LA isoforms, contributing to their complex chromatographic profiles, which included more than 10 peaks. In the specific case, HDX was based on the use of deuterated water (D_2_O) as a chromatographic solvent instead of water, allowing the H/D exchange of labile hydrogen atoms, including those located on the eventual enolic OH groups, with a 1 unit increase in the nominal molecular mass for each exchanged H atom. 

Starting from this background, the present paper will describe how HDX helped in the characterization of the multiple isoforms of OLEO and OLEA occurring in EVOO, emphasizing that the prevailing species among them were those labelled as *Open Forms II* in Figure 1, i.e., isoforms including a C=C bond between atoms C8 and C10 instead of the C8=C9 bond previously reported in the literature, thus providing unprecedented structural information on these important secoiridoids of EVOO.

## 2. Results

### 2.1. RPLC-ESI-FTMS Analysis of Oleacin and Oleocanthal in EVOO

As a first step of the structural investigation of OLEA and OLEO in EVOO extracts, eXtracted Ion Current (XIC) chromatograms related to their [M-H]^−^ ions were obtained by extracting currents using intervals centered on the corresponding exact *m*/*z* values, 319.1187 and 303.1238, respectively, and having a 0.004 *m*/*z* unit width (see Figure 2A and Figure 3A). The chromatographic features were consistent with those previously observed for both secoiridoids in other EVOO extracts [25,34,35], thus showing a qualitative reproducibility of the OLEA and OLEO profiles between different oils. Compared to those of oleuropein and ligstroside aglycones (OA and LA), usually exhibiting more than 10 peaks under the same chromatographic conditions [8,9,25,34,35], the profiles of OLEA and OLEO were significantly simpler, due to their inherently more limited structural variability. Indeed, the absence of the carboxymethylic (-CO_2_CH_3_) moiety on the C4 atom in their molecular structure (see Figure 1), with the consequent lack of an additional stereogenic center, lowers the number of potential diastereoisomers compared to those of OA and LA. Moreover, the lack of the CO_2_CH_3_ group prevents the generation of minor, closed-structure isoforms detected for OA and LA and named *Closed Forms II* [8,9]. 

As a result, the chromatographic profiles of OLEO and OLEA were dominated by just two almost identical peaks, numbered 2A and 2B in Figure 2A and Figure 3A, in apparent analogy with the chromatographic traces reported by Impellizzeri et al. [31] and Adhami et al. [32]. However, as emphasized in Figure 2A and Figure 3A, those chromatographic peaks were accompanied by some minor peaks (numbered as 1 and 3 in the case of OLEO and 1, 3A/3B and 4A/4B in the case of OLEA), which had not previously been reported. Not surprisingly, peaks related to OLEA were eluted earlier from the C18 column, i.e., between 4 and 17.5 min, than those of OLEO (between 9 and 22.5 min), likely because the additional OH group linked to the phenyl moiety increased the polarity of OLEA, reducing its interaction with the C18 stationary phase. The XIC traces of OLEA and OLEO shared other interesting features. First, the retention time difference between the two major peaks (2A and 2B), *ca*. 0.7 min, was almost constant between the two compounds and between different EVOO samples (see the XIC traces reported in Refs. [25,34,35]). Moreover, a careful comparison with the complex XIC profiles obtained for OA and LA [8,9] showed a partial similarity in terms of elution patterns, suggesting that some of the isoforms previously discovered for OA and LA were likely present, with the appropriate structural differences, also in the case of OLEA and OLEO. This analogy is highlighted in Figure 1, in which a key step for the conversion of OA/LA isoforms into OLEA/OLEO ones is represented by the methylesterases activity. Notably, four novel methylesterases were recently identified in *Olea europaea* by Volk et al. [36], exhibiting significant homology to polyneuridine aldehyde esterase (PNAE) and showing variable in vitro activity towards plant methyl esters, such as those of jasmonic acid (MeJA), indole acetic acid (MeIAA), and salicylic acid (MeSA).

Interestingly, none of those methylesterases was found to be active directly on oleuropein and ligstroside, the original secoiridoids occurring in olive drupes. However, two of them, OeEst030 and OeEst228, contributed to the transformation of OA and LA into OLEO and OLEA through demethylation, followed by non-enzymatic decarboxylation [36], as depicted in Figure 1.

The predicted structural analogies between the isomeric forms of OA and LA and those of OLEO and OLEA helped in formulating the initial hypotheses on their structures. Indeed, due to the absence of a stereogenic center on C4, two geometric isomers, related to the C=C bond located between C8 and C9, were predicted for *Open Forms I* of OLEA/OLEO, whereas a couple of diastereoisomers, arising from the combination of the fixed configuration at the chiral center on C5 with the two possible configurations assumed by the stereogenic center at C9, could be predicted for *Open Forms II* (see Figure 1). As for cyclic isomers, which correspond to *Closed Forms I* in Figure 1, up to four diastereoisomers can be expected in the case of OLEA/OLEO due to the coupling of configurations for the chiral center on C5 and the stereogenic centers on C8 and C9 (see Figure 1). 

By analogy with the chromatographic behavior observed for OA/LA isomeric forms [8,9], the retention time order for OLEA/OLEO isomers was expected to be *Open Forms I* < *Open Forms II* < *Closed Forms I*. Consequently, peaks numbered as 1 in XIC traces obtained for the two compounds (see Figure 2A and Figure 3A) were initially assigned to *Open Forms I*, peaks 2A/2B to *Open Forms II* and lately eluting peaks (3, 3A/3B and 4A/4B, according to the case) were associated with *Closed Forms I*. High-resolution MS/MS spectrometry, with or without H/D exchange, was subsequently employed to confirm these structural hypotheses and evaluate if stable enolic tautomers, able to provide further structural information, might also be present for one or more isoforms of OLEO and OLEA.

### 2.2. RPLC-ESI-FTMS/MS Analysis of Oleacin and Oleocanthal Isoforms in EVOO

Tandem MS spectra acquired using the higher energy collisional dissociation (HCD) cell of the *Q-Exactive* spectrometer, averaged in the retention time intervals corresponding to the main chromatographic peaks detected for OLEO and OLEA, are shown in panels B of Figure 2 and Figure 3, respectively. As is apparent, almost identical fragmentation patterns were obtained for the four main peaks of OLEO (1, 2A/2B, and 3), whereas two main patterns, partially overlapping, were found for OLEA, one related to peaks 1, 2A/2B, and 3B, and the other to peaks 3A, 4A, and 4B. Moreover, some common product ions were detected for isoforms of OLEO and OLEA; notably, such features were found also in the case of OA and LA isoforms [8,9], thus strengthening the structural relationship existing between the four secoiridoids.

As illustrated in the fragmentation schemes proposed for *Open Forms II* in Figure 4 and for *Open Forms I*/*Closed Forms I* in Appendix A, the key product ion to explain the fragmentation patterns observed for OLEO and OLEA isoforms was the anion of decarboxymethyl-elenolic acid (exact *m*/*z* 183.0663), which was detected in almost all MS/MS spectra, although its peak signal was more intense in the case of OLEO isoforms (see Figure 2B and Figure 3B). By analogy with the generation of deprotonated elenolic acid from the negatively charged precursors of OA and LA isoforms [8,9], the decarboxymethyl-elenolic acid anion could be generated upon gas phase collisional dissociation of any of the predicted isoforms of OLEO and OLEA shown in Figure 1. Indeed, it could be formed through the displacement of the negative charge originally located on a phenolic OH group towards one of the carbonyl groups tautomerized to enol, followed by the neutral loss of the dehydrated forms of tyrosol and hydroxytyrosol, respectively.

Additionally, structures referred to as “decarboxymethyl-elenolic acid isoforms,” arising from the predicted different isoforms for OLEO and OLEA (see Figure 1) and bearing the negative charge either on an enolic moiety or on the carboxylic one, were reported in Figure 4 and Appendix A as the starting point of the fragmentation cascades. In particular, structures including only carbonyl groups or with one of them tautomerized to a stable enol were drawn in the figure, since only keto-enolic tautomerization was able to explain the generation of some product ions once the negative charge was displaced on the enolic OH group. For example, product ions corresponding to acetate (exact *m*/*z* 59.0139), which usually corresponds to the prevailing peak signal in almost all MS/MS spectra of OLEO and OLEA, and to decarboxymethyl-elenolic acid after CO loss (exact *m*/*z* 155.0714) were reasonably generated by the all-aldehydic isoforms of decarboxymethyl-elenolic acid (see Figure 4), having the negative charge on the COOH group. On the other hand, product ions corresponding to the exact *m*/*z* ratios 165.0557, 137.0608, 123.0452, 97.0296, 95.0502, and 69.0346 could be accounted for by considering the location of the negative charge on an enolic OH group. Figure 4 and Appendix A emphasize the neutral losses of acetic acid, water, carbon monoxide, and acetylene (C_2_H_2_) involved in the gas phase formation of those product ions. Notably, a further isoform of the precursor ion, resulting from ring opening and intramolecular proton transfer and bearing the negative charge on an enolic OH functionality, had to be invoked to account for the subsequent generation of those product ions in the case of *Closed Forms I* (see Appendix A). 

As discussed so far, each of the different suggested isoforms was potentially able to generate product ions detected in the MS/MS spectra of OLEO, thus fragmentation patterns were not diagnostic for the type of isoform involved. As for OLEA, most of the product ions described so far were also detected in the tandem MS spectra referred to chromatographic peaks 1, 2A/2B, and 3B. In addition, peak signals were detected at *m*/*z* 275.1612 and 249.0766. For the sake of brevity, their generation from *Open Forms I* of OLEA is described in Appendix A, involving neutral losses of acetaldehyde or but-2-enal, respectively. However, similar mechanisms can be hypothesized to account for their generation from *Open Forms II* and *Closed Forms I* of OLEA. In the last case, a preliminary opening of the ring of decarboxymethyl-elenolic acid is required, as described for one of the pathways reported in Appendix A. 

As mentioned before, MS/MS spectra obtained for peaks 3A, 4A, and 4B in the XIC trace of OLEA suggested a partially different fragmentation pattern, as additional product ions were detected at *m*/*z* 199.0610, 181.0505, 155.0713, 139.0401, 137.0243, 111.0086, and 85.0295, with varying relative abundance (see peaks with underlined *m*/*z* ratios in Figure 3B). The interpretation of most of these product ions was already provided in one of our previous studies, concerning the evolution of secoiridoids in EVOO stored for a long time (up to six months) in the presence of oxygen [25]. Indeed, as described in Appendix A, the *m*/*z* ratio (319.1187) of monoisotopic deprotonated OLEA [C_17_H_19_O_6_]^−^, is identical to that of deprotonated oleocanthalic acid, i.e., the anion of oleocanthal in which a carbonyl group has been oxidized to a COOH group. All possible isoforms of oleocanthalic acid are reported in Appendix A, depending on the OLEO isoform involved in the oxidation process and on the final location of the COOH group (in fact, *Open Forms I* and *II* of OLEO have two C=O groups that can undergo oxidation to COOH). As shown in the figure, the product ion detected at *m*/*z* 199.0610, prevailing over others in the MS/MS spectra of peaks 3A, 4A, and 4B shown in Figure 3B, corresponds to deprotonated oleocanthalic acid after the neutral loss of the dehydrated form of tyrosol (exact *m*/*z* 199.0612), a typical fragmentation of secoiridoids. Moreover, as shown in Appendix A for *Closed Forms I* of oleocanthalic acid, the *m*/*z* 199.0612 ion is the precursor of a fragmentation cascade involving neutral losses of acetic acid, water, acetylene, carbon dioxide, and ethylene, and is, according to the case, able to account for the entire series of additional product ions mentioned above.

Based on these considerations, OLEA peaks labelled 3A, 4A, and 4B in Figure 3A could be associated with oleocanthalic acid isoforms, formed upon EVOO oxidation during storage before analysis. As recently discussed [25], their retention times suggest that these peaks are related to diastereoisomers of *Closed Forms I* of oleocanthalic acid; due to the presence of two stereogenic centers (at C8 and C9), each with two possible configurations, and to the fixed configuration of the chiral center at C5, four isoforms are expected for them. Actually, other isoforms of oleocanthalic acid, corresponding to *Open Forms II*, were detected during our previous investigation, but only after prolonged (i.e., after six months) EVOO storage [25], and their retention times were much lower than those observed in the present case. It is worth noting that the proximity of peak 3B, related to one of the OLEA isoforms, with peaks 3A and 4A, associated with isoforms of oleocanthalic acid, may explain why the MS/MS spectrum of the latter also included product ions typical of OLEA isoforms (i.e., those detected at *m*/*z* 183.0660 and 165.0555), due to spectral interference. On the other hand, as evidenced in Appendix A, product ions with *m*/*z* ratios compatible with exact values 59.0139 (the acetate ion), 95.0502 and 69.0348, typical of OLEO and OLEA isoforms (see Figure 4 and Appendix A), can be generated also by isoforms of oleocanthalic acid, thus they were detected also in the MS/MS spectra related to peaks 3A and 4A of OLEA (see Figure 3B). Actually, their peak intensity was negligible only in the MS/MS spectrum related to the oleocanthalic acid isoform corresponding to peak 4B, for which the pathway leading to the *m*/*z* 111.0088 ion was clearly prevailing. Since the key fragmentation towards the generation of this ion was the neutral loss of acetic acid, with the formation of a C=C bond between C5 and C9 (see Appendix A), peak 4B might correspond to the specific diastereoisomer of oleocanthalic acid bearing the H atom bond to C9 on the same side as the CH_2_COOH group bond to C5, thus making the detachment of acetic acid more sterically favorable.

Notably, the same results described so far for OLEO and OLEA extracted from the EVOO sample were obtained by applying the RPLC-ESI-(-)-FTMS method to a mixture of OLEO and OLEA standards available commercially (phyproof^®^ Reference Substances), each dissolved at a 10 mg/L concentration into the same water/methanol mixture used for the secoiridoid extraction from EVOO. Indeed, as highlighted in Appendix A, the profiles inferred from the XIC chromatograms of the two standard compounds were very similar to those obtained from the real EVOO sample, the only relevant exception being the absence of peaks related to oxidized oleocanthal, as expected. Even in this case, the MS/MS spectra obtained for the different detected isoforms of OLEO and OLEA were very similar.

Consequently, despite the ability to provide some structural information on OLEO and OLEA isoforms and also on oleocanthalic acid isoforms, tandem MS analyses were unable to associate each chromatographic peak to specific isoforms of OLEO and OLEA among those depicted in Figure 1. RPLC-ESI-FTMS analysis in conjunction with H/D exchange was thus undertaken to address this issue.

### 2.3. RPLC-ESI-FTMS and MS/MS Analysis of Oleacin and Oleocanthal Isoforms in the Presence of H/D Exchange

As previously described [8,9], H/D exchange (HDX) proved a powerful approach to detect the existence of stable dienolic tautomers for some isoforms of oleuropein and ligstroside aglycones, namely *Open Forms II*, providing fundamental information to recognize them. HDX was performed by using D_2_O as the co-solvent of acetonitrile in the RPLC mobile phase. As in the case of OA and LA, labile hydrogen atoms of phenolic and enolic -OH groups of OLEO and OLEA were expected to be involved in the H/D exchange. Thanks to the high mass resolving power available with the Orbitrap analyzer of the *Q-Exactive* spectrometer, the *m*/*z* shift related to the exchange of an H atom with a D one can be recognized distinctly with respect to the one related to the exchange of a ^12^C atom with a ^13^C one, the main responsible for the *m*/*z* differences occurring between the M+0 and M+1 isotopologues of the secoiridoids under study. As an example, the exact *m*/*z* value for the M+1 isotopologue of deprotonated OLEO, 304.1272, could be distinguished from the one related to the M+0 isotopologue of oleocanthal after one H/D exchange, 304.1301, even though they only differed by 0.0029 *m*/*z* units. Appropriate *m*/*z* intervals, having a width of 0.004 units and including the exact *m*/*z* values predicted for the M+0 isotopologues of non-deuterated and of the possible deuterated forms of OLEO and OLEA ions, could thus be adopted to retrieve the respective extracted ion current (XIC) traces from RPLC-ESI(-)-FTMS TIC chromatograms obtained when using D_2_O in the mobile phase, as shown in Figure 5 and Figure 6, respectively. As was apparent, mono-, bis-, or tris-deuterated OLEO and OLEA ions were detected, along with non-deuterated ions, resulting from the total lack of H/D exchange concerning part of the ionic population. Tetra-deuterated ions were also detected for OLEA. As described in detail in Table 1, all possible isoforms of OLEA and OLEO reported in Figure 1, along with those arising from them as a result of keto-enolic tautomerism on carbonyl groups, and all the reasonable mechanisms for H/D exchange, including one that was initially not expected (*vide infra*), had to be considered to explain the experimental data. As discussed for OA and LA [8,9], the H atoms located on the phenolic OH group(s) of OLEO and OLEA, along with those included in enolic OH groups, were considered the first candidates for the HDX process.

Starting from dialdehydic *Open Forms I* and *II* and mono-aldehydic *Closed Forms I* of OLEO (see the first row in the related part of Table 1), the single phenolic OH group is the only predictable site for H/D exchange, but it is also the site of negative ionization. Consequently, if the phenolic OH group is involved in the H/D exchange, the D atom is subsequently lost as a D^+^ ion during the ESI process, and no deuteration can be finally observed for negatively charged OLEO (exact *m*/*z* value: 303.1238). Conversely, if the enolization of one of the suitable aldehydic groups occurs, the H atom of the resulting enolic OH group can be exchanged with D, and a negative ion with an exact *m*/*z* of 304.1301 is formed (see the second row in the OLEO section of Table 1). It is thus not surprising that peaks for all possible OLEO isoforms were detected in the XIC chromatograms referred to exact *m*/*z* values 303.1238 and 304.1301 (see Figure 5A,B), since all of them could include none or at least one D atom in the final ion. Notably, since peaks in the two XIC traces were well aligned in terms of retention times, the enolization leading to a D atom introduction in the deprotonated OLEO structure reasonably occurred during the ionization process in the ESI source.

As evidenced in Figure 5C, isoforms of deprotonated OLEO including two D atoms on their structure (exact *m*/*z* 305.1364) were also detected, and, once again, the corresponding chromatographic peaks were perfectly aligned with those of non- and mono-deuterated anions. In principle, a double HDX on labile H atoms should have been possible only for *Open Forms II*, since only they can undergo a double keto-enolic tautomerization, leading to the presence of two enolic OH groups on their structure. However, the XIC trace related to the *m*/*z* value 305.1364 (see Figure 5C) showed all four peaks expected if all OLEO isoforms were able to generate anions with two D atoms on their structures. An interesting possible explanation for this surprising outcome was obtained by considering also the deuteration of specific non-labile H atoms, i.e., those located in the α-position to carbonyl groups. Indeed, Zherebker et al. [37] disclosed that a relatively high desolvation temperature can facilitate the H/D exchange at non-labile sites of hydroxybenzoic acids and aromatic amino acids during the ESI process. According to this model, schematically described in Figure 7, for dialdehydic *Open Forms I* of OLEO and OLEA, an additional H/D exchange would be induced by the presence of deuterated hydroxide ions in the microdroplets generated during the ESI process in negative polarity. Remarkably, the acceleration of organic reactions in the confined volume of partially solvated microdroplets formed in an ESI source was discussed in a review by Yan et al. [38]. In the case of OLEO *Open Forms I*, an additional deuteration would result from the in-source H/D exchange at C4, along with that occurring on the enolic OH group linked to C3, as shown in the third row of the OLEO-related part of Table 1. As emphasised in Figure 7, the deuterated enol with a D also on C4 can be considered in equilibrium with the tautomer bearing a carbonyl group on C3 and two D linked to C4 (only the former structure was reported in Table 1, for the sake of simplicity). In the case of OLEO *Open Forms II*, the described double deuteration mechanism described in Figure 7 might involve, alternatively, the other C=O group, since this has one H atom on the α-carbon (C9) that can be exchanged with D. As for *Closed Forms I*, only a ring opening process occurring during the ESI process, such as the one already described as part of a fragmentation pathway reported in Appendix A, would finally result in the presence of two enolic groups and, hence, in the detection of OLEO anions including 2 D atoms (see the third row in the upper part of Table 1), thus explaining why peaks for those isoforms were detected also in the XIC trace referred to the *m*/*z* value 305.1364 (see Figure 5C). Notably, all double H/D exchanges described for OLEO isoforms were expected to occur during the ESI process because the corresponding peaks detected in XIC chromatograms of non-, mono-, and bis-deuterated OLEO anions (see Figure 5A–C) were aligned.

A different scenario was hypothesized to explain the XIC chromatogram obtained for the exact *m*/*z* value of 306.1426, which corresponds to OLEO anions bearing 3D atoms on their structure. As shown in Figure 5D, no relevant peak due to *Open Forms I* was detected since there was no reasonable way for these forms to undergo three H/D exchanges. However, four peaks were found in the retention time interval related to *Closed Forms I*, one of which (3A, at 22.20 min) aligned with the prevailing peak detected in that interval in other XIC traces of deuterated OLEO derivatives. In this case, the already cited ring opening of decarboxymethyl-elenolic acid enabled the third H/D exchange to occur at C4 (see the last row in the upper part of Table 1), through the mechanism described in Figure 7. Interestingly, this process seemed to occur with a non-negligible likelihood for all four possible diastereoisomers predicted for *Closed Forms I*, since three weak additional chromatographic peaks (labeled 3B, 3C, and 3D in Figure 5D) were detected along with the prevailing peak 3A.

The detection of four distinct peaks in the XIC chromatogram referred to tris-deuterated OLEO anions instead of the two peaks (2A/2B) putatively associated with *Open Forms II* required a specific explanation. In particular, couples of peaks located on both sides of them (thus not aligned with them), labelled as 2A’/2A’’ and 2B’/2B’’ (see Figure 5D), were detected. As emphasized in the last row of the OLEO-related part of Table 1, these peaks can be associated only with OLEO isoforms in which a double H/D exchange occurs on the enolic moiety involving C3 and C4, such as the one already described for *Open Forms I*, and an additional H/D exchange occurs on the enolic group resulting from the tautomerization on the carbonyl group involving C1. Such an extent of deuteration is possible only for *Open Forms II*, since only in their case the C^1^=O group can be involved in the keto-enolic tautomerization. The detection of peaks at different retention times compared to those of the two typical peaks observed (2A/2B) suggests that dienolic forms of *Open Forms II* can be formed in the mobile phase and be so stable as to be separated from their all-aldehydic counterparts. Those forms likely underwent the first two H/D exchanges on their enolic OH groups, whereas the last H/D exchange, on C4, occurred during the ESI process. This behavior is analogous to the one previously observed for the *Open Forms II* of oleuropein and ligstroside aglycones [8,9] and can be considered an important confirmation of the structural characteristics of these isoforms (namely, the occurrence of a C=C bond between C8 and C10) that cannot be obtained otherwise using MS. Strikingly, the detection of four distinct peaks for the tris-deuterated derivatives of OLEO *Open Forms II*, with no alignment with peaks 2A/2B, is consistent with the occurrence of dienolic forms, in which all the combinations of the geometries (*E* or *Z*) of the two C=C bonds related to the enolic moieties are possible, as emphasized in Table 1 by a specific drawing of bonds linking the C3 and C1 atoms to the enolic OH groups.

As evidenced by Figure 6, OLEA exhibited a behavior similar to that of OLEO when the H/D exchange was performed. In this case, since an additional OH not involved in negative ionization is present on the catechol ring, an additional H/D exchange occurred with respect to OLEO. XIC chromatograms for mono-, bis-, and tris-deuterated isoforms, corresponding to non-, mono-, and bis-deuterated isoforms of OLEO, respectively, showed peaks aligned with those observed for non-deuterated isoforms of OLEA. As shown in the lower part of Table 1, the first deuteration could be expected to occur on the OH group linked to the phenyl ring. The second deuteration was still possible for all isoforms since all of them had a carbonyl group suitable for the keto-enolic tautomerization in the ESI source, enabling a H/D exchange to occur on an enolic OH group (see the third row in the lower part of Table 1). The third deuteration required the same mechanism described before for the double deuteration on OLEO isoforms (see Figure 7) when *Open Forms I* and *II* of OLEA were involved, whereas a ring opening was essential to enable a further H/D exchange in *Closed Forms I* of OLEA (see structures reported in the fourth row of the OLEA-related part of Table 1). The alignment of corresponding peaks in XIC traces referred to non-, mono-, bis-, and tris-deuterated OLEA isoforms was observed (see Figure 6A–D), by analogy with OLEO isoforms. Once again, a specific behavior, similar to the one found for tris-deuterated OLEO, was observed when the XIC trace referring to tetra-deuterated OLEA anions was considered (see Figure 6E). Indeed, while no significant peak was observed for *Open Forms I* and an alignment with the corresponding peak in the other traces (peak 3B) was found for *Closed Forms I*, two couples of peaks were detected instead of peaks 2A and 2B for *Open Forms II* that were not aligned with them. This outcome confirmed the occurrence of stable dienolic forms, with distinct retention times and capability to undergo H/D exchange already in the mobile phase, for *Open Forms II*, thus providing an indirect but important proof of the specific structural features of these isoforms, prevailing over others also in the case of oleacin.

A final consideration is due to the peak labeled 4B, detected after *ca.* 17 min in four of the five XIC traces reported in Figure 6. As discussed before, that peak was related to one of the *Closed Forms I* of oleocanthalic acid, eluting near the *Closed Forms I* of oleacin (see Figure 3). As it can be inferred by considering the structure of oleocanthalic acid *Closed Forms I* (see Appendix A), these isoforms can undergo one H/D exchange on the OH group linked to the phenyl ring, assuming the carboxylic group as the most reasonable ionization site in this case. One additional H/D exchange can occur on the enolic group linked to C3, resulting from the ring opening occurring during the ESI process and described before for *Closed Forms I* of OLEO and OLEA. Finally, a H/D exchange may occur on the C9 atom, which is an α-carbon with respect to the COOH group and thus can participate in the mechanism described in Figure 7. Accordingly, a peak corresponding to peak 4B in the XIC trace of non-deuterated OLEA was observed in traces referred to as mono-, bis-, and tris-deuterated species (see Figure 6B–D), whereas there was no reasonable possibility for oleocanthalic acid *Closed Forms I* to undergo four H/D exchanges, and thus no significant peak was detected at the retention time of peak 4B in the XIC trace for tetra-deuterated OLEA (see Figure 6E).

## 3. Discussion

The structural investigation of the isomeric forms of oleocanthal and oleacin in extra-virgin olive oil confirmed the challenges posed by other EVOO secoiridoids, i.e., the aglycones of oleuropein and ligstroside [8,9], with multiple peaks detected in RPLC-ESI-(-)-FTMS chromatograms and almost identical fragmentation patterns inferred from the corresponding MS/MS spectra. MS/MS analyses enabled only the distinction between peaks related to oxidized forms of oleocanthal and those related to oleacin isoforms, which were detected in the same extracted ion current chromatogram since the two compounds exhibit the same molecular weight. 

The use of HDX exchange in conjunction with RPLC-ESI-FTMS provided fundamental information to clarify the structural features of OLEO and OLEA, suggesting that the two main isoforms of these secoiridoids corresponded to *Open Forms II* shown in Figure 1, i.e., isoforms with a C=C bond located between C8 and C10 and a stereogenic center on C9, determining the occurrence of a couple of diastereoisomers (through coupling with the chiral center on C5). Indeed, only these structural features were consistent with the generation, already in the RPLC mobile phase, of four stable dienolic tautomers exhibiting distinct retention times and subsequently detected as tris- or tetra-deuterated ions, in the cases of OLEO and OLEA, respectively (see Table 1). 

Notably, one of the few studies reported so far in which two resolved chromatographic peaks were detected for oleocanthal and NMR was also adopted, along with MS, for structural characterization [32], indicated the C=C bond to be located between the C8 and C9 atoms, such as in *Open Forms I* shown in Figure 1. In fact, ^1^H NMR spectra were consistent with the presence of three H atoms linked to C10 and one H atom linked to a vinylic carbon, i.e., to C8 [32]. A possible explanation for the discrepancy observed between this study and the one in Ref. [32] might be found considering the aglycones of oleuropein and ligstroside, which are the most likely precursors of OLEO and OLEA, as explained before (and as hypothesised already in early studies, such as Ref. [11]). Indeed, EVOOs rich in *Open Forms II* of OA and LA, such as the one employed during the present study (see Appendix A for XIC traces referred to the two secoiridoids), are expected to contain relevant concentrations also of the corresponding isoforms of OLEA and OLEO. This would not be the case of EVOOs in which *Open Forms I* of OA and LA were prevailing; one of them might have been analyzed in Ref. [32]. The outcome of an experiment mimicking the generation of OA isoforms during olive oil production, performed in our laboratory through a reaction of oleuropein with β-glucosidase, followed by incubation at acid pH [8], might provide an additional explanation. Indeed, the experiment showed that *Closed Forms I* of OA were initially generated from oleuropein, upon detachment of glucose catalyzed by β-glucosidase, then a partial transformation into *Open Forms I* occurred upon acidification. Only incubation at acid pH for several days led to the appearance of the first isoforms of *Open Forms II* [8]. The time elapsed between EVOO production and the extraction/analysis of secoiridoids, including OLEO and OLEA, might thus play a relevant role in determining the prevailing isoforms observed. In any case, the analytical method described in the present paper appears suitable to assess the distribution of the different isoforms of the two secoiridoids in a specific EVOO, analyzed either soon after production or after a specific storage time.

## 4. Materials and Methods

### 4.1. Chemicals and Olive Oil Samples

Water, methanol (LC-MS grade), hexane (HPLC-grade), and oleuropein (i.e., 2-(3,4-dihydroxyphenyl)ethyl(2S-(2α,3E,4β))-3-ethylidene-2-(β-D-glucopyranosyloxy)-3,4-dihydro-5-(methoxycarbonyl)-2H-pyran-4-acetate), deuterated water (isotopic purity higher than 99.99%), and standards of oleocanthal and oleacin (phyproof^®^ Reference Substances) were purchased from Merck/Sigma-Aldrich (Milan, Italy). 

An Italian extra-virgin olive oil (EVOO) produced in the Apulia region of Italy using a blend of olives from two cultivars (70% Coratina, 30% Ogliarola) was selected for the extraction and subsequent characterization of oleocanthal and oleacin isoforms. A further Apulian EVOO, produced using a single olive cultivar (100% Coratina), was also considered for a test of OLEA/OLEO extraction efficiency. To minimize the occurrence of oxidative degradation of oleocanthal, which, as explained before, could complicate the profile of oleacin, the oils were subjected to secoiridoid extraction soon after the respective bottle opening.

### 4.2. Extraction of Secoiridoids from Olive Oil

The extraction of oleocanthal and oleacin from the olive oil was performed using a CH_3_OH/H_2_O 60:40 (*v*/*v*) mixture, recently employed also for the analysis of oleuropein and ligstroside aglycones [8,9]. Specifically, 2 g of extra virgin olive oil were dissolved in 3 mL of HPLC-grade hexane, and the resulting solution was vortexed for 1 min. Afterwards, 500 μL of the CH_3_OH/H_2_O 60:40 (*v*/*v*) extracting solvent were added, and the resulting mixture was vortexed for 2 min and then sonicated for 4 min to facilitate the extraction of oil polar compounds (including secoiridoids) in the methanolic-aqueous phase. After centrifugation at 4000 rpm (corresponding to a centrifugal acceleration of 2000× *g*), the latter was separated from the hexane-rich phase, withdrawn with a microsyringe, and stored in a glass tube. The hexane-rich phase was used for a further extraction with 500 μL of the extracting solvent; the resulting two aliquots of methanolic/aqueous phase were pooled and washed for 1 min with 2 mL of hexane under vortexation, to remove eventually dissolved residual apolar compounds. After further centrifugation at 4000 rpm for 5 min, the methanolic-aqueous extract was removed and transferred into a glass vial, which was closed with a screw cap after saturating its headspace with nitrogen and stored at +4 °C to minimize the eventual oxidation of extracted secoiridoids before LC-MS analysis.

The extraction efficiency of the described procedure towards oleocanthal and oleacin was evaluated using the same procedure adopted for the aglycones of oleuropein and ligstroside [8,9], performed on the two different EVOOs cited in Section 4.1. In particular, a 2 g aliquot of each oil was subjected to three extraction steps: the first was based on two 500 μL aliquots of the CH_3_OH/H_2_O 60:40 (*v*/*v*) mixture, finally pooled and analyzed by RPLC-ESI-FTMS. Two subsequent, separate extraction steps, each based on a 500 μL aliquot of the extraction solvent, were performed. Each of the three resulting extracts was fortified with oleuropein at a 100 mg/L concentration and used as an internal standard. Indeed, the peak areas obtained from the XIC chromatograms related to oleacin and oleocanthal after analyzing each extract by RPLC-ESI-FTMS in negative polarity (see Section 4.3 for details on ion current extraction), were ratioed to that obtained for oleuropein. The resulting ratios were considered to be values proportional to the concentrations of extracted secoiridoids, as instrumental response fluctuations were minimized through normalization to the internal standard. The ratios were thus used to estimate the extraction yield at each step of the procedure (with differences in solvent volume referred to the two final extracts, compared to the first one, accounted for in the calculation). The entire procedure was performed in duplicate for each of the two oils considered, and average yields of 97 and 95% were already estimated after the first extraction step for oleacin and oleocanthal, respectively, with differences between the two olive oils not exceeding 2%. These results were in excellent accordance with the average extraction yields (95%) obtained for oleuropein aglycone in the same oils [8] and confirmed the effectiveness of the adopted methanol/water mixture in extracting secoiridoids from olive oils.

### 4.3. RPLC-ESI-FTMS Instrumentation and Operating Conditions

RPLC-ESI-FTMS analyses on oleocanthal and oleacin were performed using an Ultimate 3000 UHPLC system coupled to a *Q-Exactive* quadrupole-Orbitrap mass spectrometer (Thermo Scientific, Waltham, MA, USA). RPLC separations were based on an Ascentis Xpress C18 column (150 × 2.1 mm ID, 2.7 µm particle size) preceded by an Ascentis Xpress C18 (5 × 2.1 mm ID) security guard cartridge (Supelco). Olive oil extracts (5.0 μL volume) were injected using the autosampler included in the Ultimate 3000 UHPLC system, equipped with a 6-way Rheodyne valve. 

The following elution gradient, based on water (solvent A) and acetonitrile (solvent B), was recently adopted successfully for the separation of oleuropein/ligstroside aglycone isoforms in olive oil extracts [8,9], was adopted: [ 0–5 min) 20% solvent B; 5–35 min) from 20% to 50% (*v*/*v*) solvent B; 35–40 min) from 50% to 100% of solvent B; 40–50 min) isocratic at 100% solvent B; 50–55 min) from 100% to 20% [ of solvent B; 55–70 min) column reconditioning at 20% of solvent B. HDX experiments were performed by replacing water with deuterated water during RPLC-ESI-FTMS analyses. The flow rate was always set at 200 μL/min and the column temperature at 25 °C. As recently discussed [8,38], no acidifying additive was introduced in the mobile phase since they were found to alter the profile of secoiridoids originally extracted from olive oils. The effect is particularly enhanced in the case of oleocanthal and oleacin [38]. On the other hand, no attempt was made to increase the mobile phase pH to facilitate the negative ionization of the two secoiridoids since, by analogy with oleuropein [39], the pKa value(s) of their phenolic OH groups were expected to be higher than 9 [40]. Nonetheless, similarly to oleuropein and ligstroside aglycones [8,9], oleocanthal and oleacin were easily turned into the corresponding [M-H]^−^ ions during the ESI process, through deprotonation of the single or of one of the two phenolic OH groups available, thus enabling their MS detection in negative ion mode with a good signal/noise ratio. 

ESI(-)-FTMS *full scan* acquisitions were performed, in the *m*/*z* range 200–2000, after setting the main parameters of the heated ESI (HESI) interface and of the ion optics of the Q-Exactive spectrometer as follows: sheath gas flow rate, 60 (arbitrary units); auxiliary gas flow rate, 15 (arbitrary units); spray voltage, −4 kV; capillary temperature, 200 °C; S-lens RF level, 100 (arbitrary units). Precursor ions for FTMS/MS analyses, corresponding to the first isotopologue of the [M-H]^−^ ion of each secoiridoid, were preliminarily isolated in the quadrupole analyzer of the *Q-Exactive* spectrometer and then fragmented into its higher energy collisional dissociation (HCD) cell at a normalized collisional energy (NCE) of 20 (a.u.). This energy led to the generation of several product ions, thus providing interesting structural information for both investigated secoiridoids. The *Q-Exactive* spectrometer was calibrated daily by the infusion of a calibration solution for negative polarity measurements, provided by the instrument manufacturer. As a result, a mass accuracy better than 5 ppm was usually achieved. The LC-MS instrumentation was controlled by the Xcalibur software (Thermo Scientific), which was also used for ion current extraction and focused on the main isotopologue of the [M-H]^−^ ion of each secoiridoid (extraction window width: 0.004 *m*/*z* units). The ChemDraw Pro 8.0.3 software (CambridgeSoft Corporation, Cambridge, MA, USA) was employed to draw chemical structures.

## 5. Conclusions

The implementation of H/D exchange (HDX) in the RPLC-ESI(-)-FTMS analysis of oleocanthal and oleacin extracted from an EVOO sample gave the opportunity to clarify the complex chromatographic profiles exhibited by the two secoiridoids, consisting of at least four different peaks (not considering those referred to oleocanthalic acid isoforms, appearing in the same trace of oleacin, due to the identity of molecular masses). In particular addition, the maximum extent of deuteration observed for stable dienolic tautomers formed by the two prevailing isoforms of both compounds indicated the latter as an open-structure dialdehydic species including a C=C bond in a different position (C8–C10, according to the atom numbering usually adopted for secoiridoids) than the one (C8–C9) usually hypothesized for them, based on NMR spectroscopy. This feature was consistent with that previously observed, using the same approach, for some major isoforms of the aglycones of oleuropein and ligstroside, which are the most likely precursors of oleacin and oleocanthal in EVOO, respectively. As discussed in the paper, the prevalence of isoforms with one of the two cited structural features in a specific olive oil might depend on the cultivar adopted for production and also on the time elapsed between production and analysis, since processes transforming one of them into the other might occur upon olive oil storage.

Assessing whether the new structural feature found for oleacin and oleocanthal isoforms might play a role in explaining the peculiar bioactivity observed for these secoiridoids would be an important additional step of the present study. Indeed, a complete understanding of the structure-bioactivity relationship related to EVOO secoiridoids might pave the way to the development of novel and more effective anti-inflammatory drugs mimicking such natural chemical structures.

## Figures and Tables

**Figure 1 molecules-28-02066-f001:**
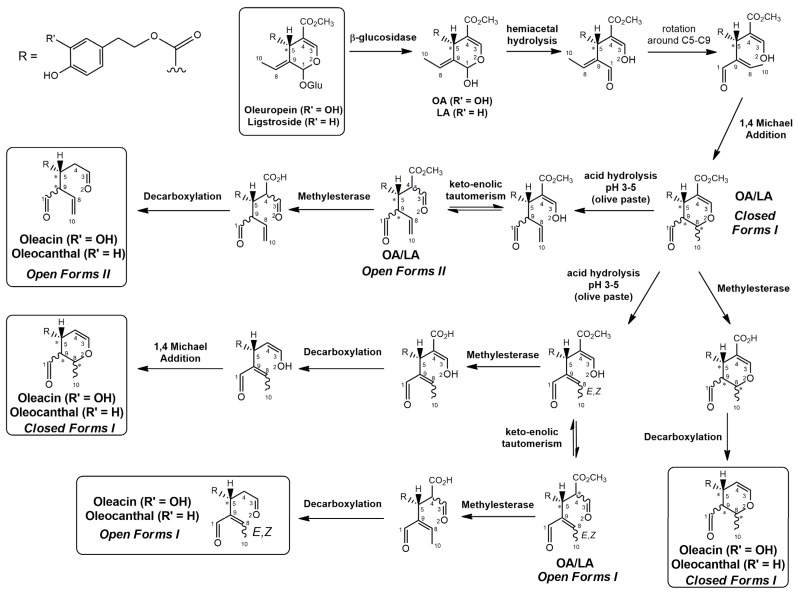
Schematic representation of the complex sequence of enzymatic and chemical processes occurring during olive oil production, leading from the main secoiridoids contained in olive drupes, oleuropein and ligstroside, to oleuropein and ligstroside aglycones (OA/LA) and finally to oleocanthal and oleacin, whose main isoforms are shown. See text for details. The chiral center on C5 and the stereogenic center on C9 are marked by an asterisk.

**Figure 2 molecules-28-02066-f002:**
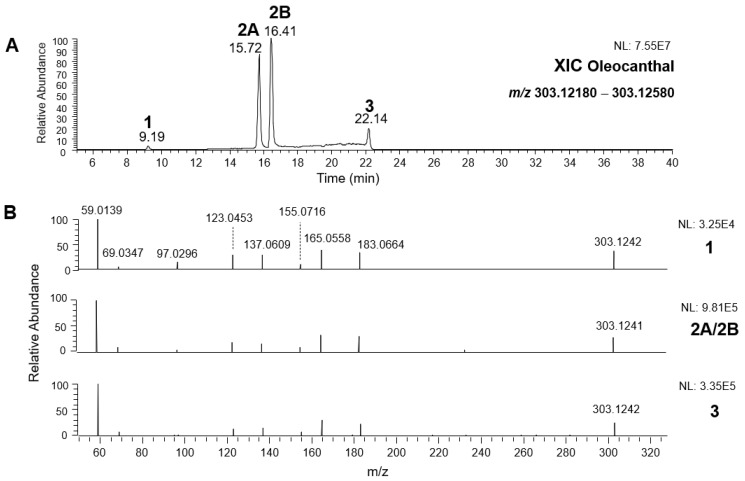
(**A**) Extracted Ion Current (XIC) chromatogram referred to monoisotopic deprotonated OLEO ([M-H]^−^, *m*/*z* 303.1238) obtained after the RPLC-ESI(-)-FTMS analysis of an EVOO extract. (**B**) HCD-FTMS/MS spectra averaged under the main chromatographic peaks detected in the XIC trace. See Figure 4 and Appendix A for structures hypothesized for each product ion. To avoid redundancy, *m*/*z* labels for product ions, identical in all cases, were reported only in the first spectrum. NL = Normalization Level.

**Figure 3 molecules-28-02066-f003:**
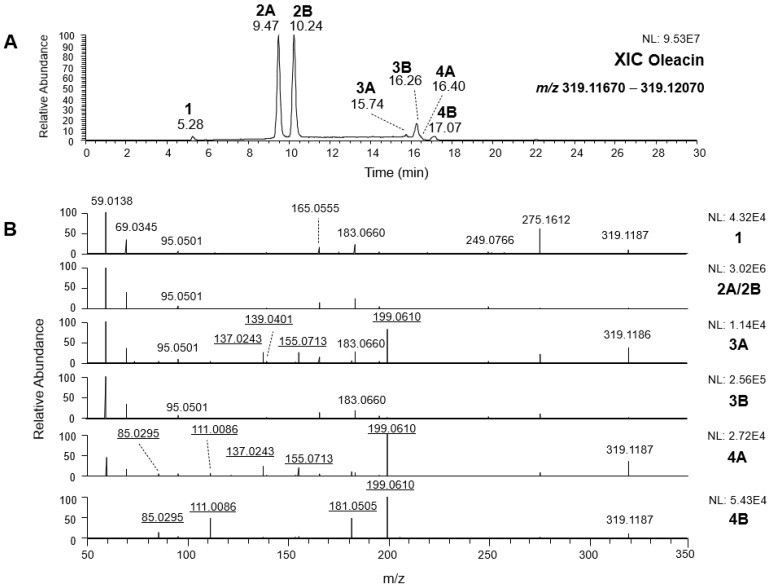
(**A**) Extracted Ion Current (XIC) chromatogram referred to monoisotopic deprotonated OLEA ([M-H]^−^, *m*/*z* 319.1187) obtained after the RPLC-ESI(-)-FTMS analysis of an EVOO extract. (**B**) HCD-FTMS/MS spectra averaged under the main chromatographic peaks detected in the XIC trace. See Figure 4 and Appendix A for structures hypothesized for each product ion detected. Underlined *m*/*z* ratios are referred to product ions related to oleocanthalic acid isoforms. To avoid redundancy, *m*/*z* labels for product ions identical in different spectra were not reported in all spectra. NL = Normalization Level.

**Figure 4 molecules-28-02066-f004:**
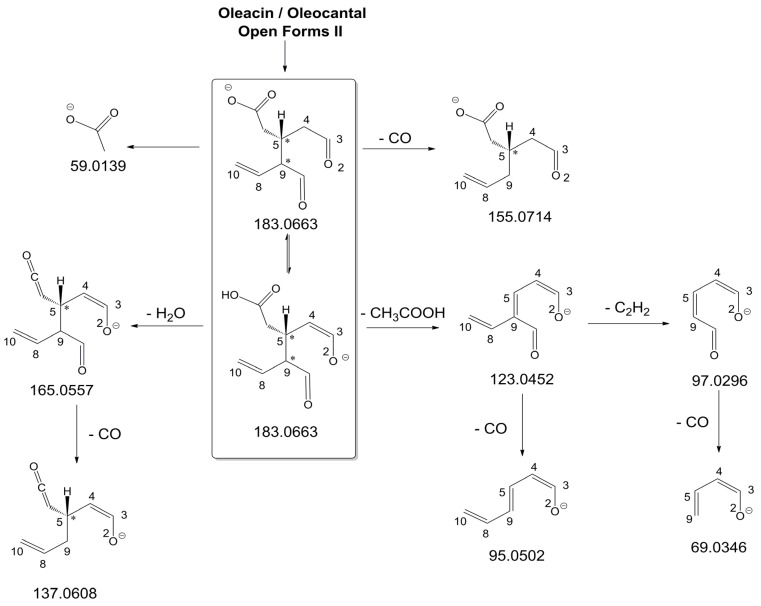
Fragmentation pathways hypothesised to explain the generation of product ions detected in MS/MS spectra referred to oleacin and oleocanthal chromatographic peaks (see Figure 2B and Figure 3B), starting from their common leading product ion, i.e., the anion of decarboxymethyl-elenolic acid (exact *m*/*z* 183.0663). Pathways proposed for *Open Forms II* hypothesised for the two secoiridoids (see Figure 1) are shown (see Appendix A for those proposed for *Open Forms I* and *Closed Forms I*). The chiral center on C5 and the stereogenic center on C9 are marked by an asterisk.

**Figure 5 molecules-28-02066-f005:**
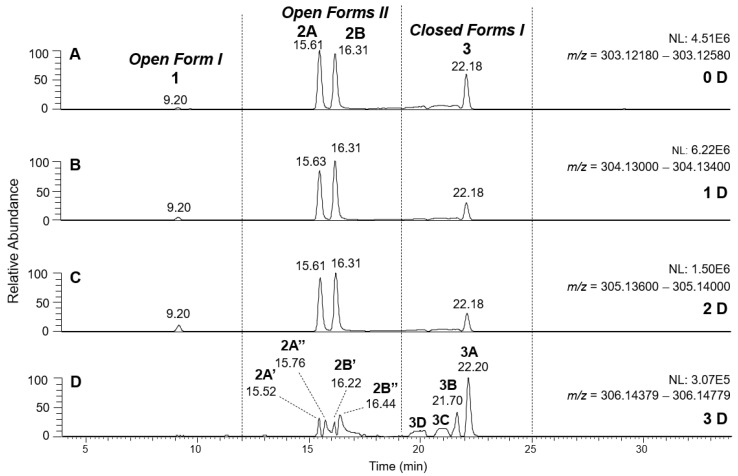
Comparison between XIC chromatograms obtained for the anions of non-deuterated (0 D, panel (**A**)) and mono- (1 D, panel (**B**)), bis- (2 D, panel (**C**)) and tris- (3 D, panel (**D**)) deuterated isoforms of OLEO, resulting from the HDX occurring during the RPLC-ESI(-)-FTMS analysis of an EVOO polar extract using D_2_O as co-solvent of acetonitrile in the mobile phase. The *m*/*z* intervals adopted for the extraction of ion currents are reported along with the normalization levels (NL). The proposed structure for each isoform is depicted in Table 1.

**Figure 6 molecules-28-02066-f006:**
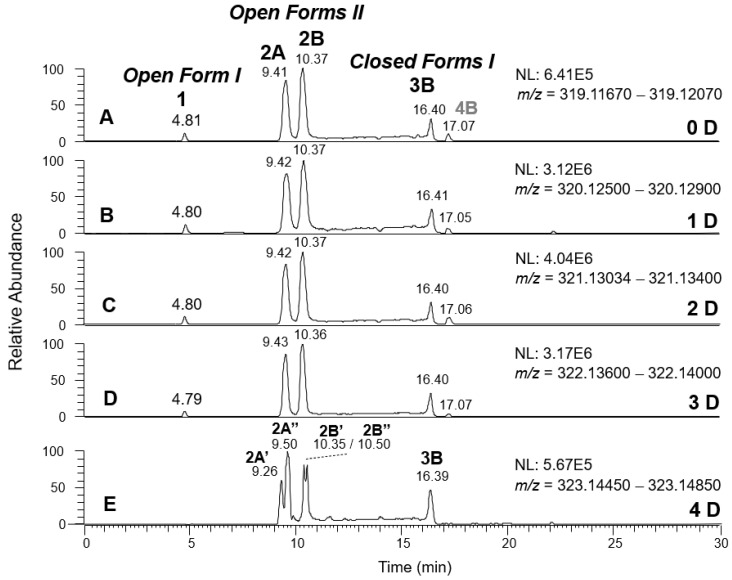
Comparison between XIC chromatograms obtained for the anions of non-deuterated (0 D, panel (**A**)) and mono- (1 D, panel (**B**)), bis- (2 D, panel (**C**)), tris- (3 D, panel (**D**)) and tetra- (4 D, panel (**E**)) deuterated isoforms of OLEA, resulting from the HDX occurring during the RPLC-ESI(-)-FTMS analysis of an EVOO polar extract using D_2_O as co-solvent of acetonitrile in the mobile phase. The *m*/*z* intervals adopted for the extraction of ion currents are reported along with the normalization levels (NL). The proposed structure for each isoform is depicted in Table 1. Peak 4B corresponds to the major isoform of oleocanthalic acid, formed upon oxidative degradation of OLEO and isomeric with OLEA (see text for details).

**Figure 7 molecules-28-02066-f007:**
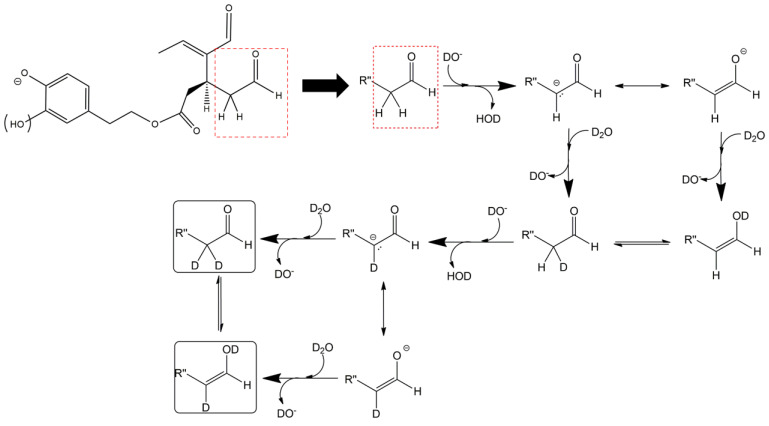
Proposed mechanism explaining the double HDX occurring during the ESI process on the carbon atom(s) adjacent to carbonyl groups of OLEO and OLEA isoforms and not involved in a C=C bond. For the sake of simplicity, the mechanism is shown only for the suitable C=O group present in *Open Forms I* of the two secoiridoids, although it can also occur on all suitable carbonyl groups of *Open Forms II* and *Closed Forms I* (see Table 1). The existence of a tautomeric equilibrium between bis-deuterated aldehydic forms and their enolic counterparts is highlighted.

**Table 1 molecules-28-02066-t001:** Summary of the suggested structures for the anions of non-deuterated oleocanthal (OLEO) and oleacin (OLEA) isoforms and for the corresponding deuterated derivatives detected upon H/D exchange during RPLC-ESI-FTMS analysis of an EVOO extract. The number of D atoms introduced in each case and the exact *m*/*z* value of the corresponding M+0 isotopologue are reported in the first column. Isoform numerical labels are the same adopted in Figure 5 and Figure 6 for OLEO and OLEA, respectively. The chiral center on C5 and the stereogenic center on C9 are marked by an asterisk.

D Atoms and Exact *m/z* Value (M+0)	*Open Forms I*	*Open Forms II*	*Closed Forms I*
Deprotonated OLEO, [C_17_H_19_O_5_]^−^			
0 D (303.1238)	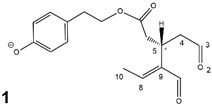	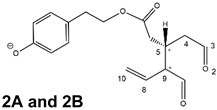	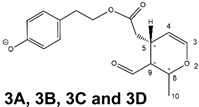
1 D (304.1301)	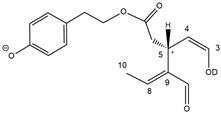	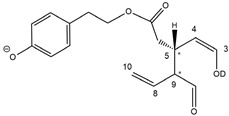	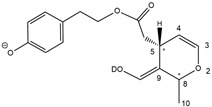
2 D (305.1364)	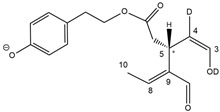	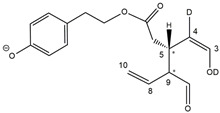	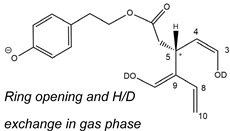
3 D (306.1426)		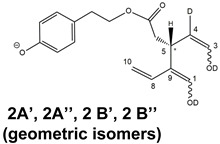	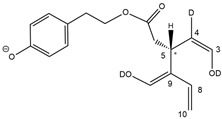
Deprotonated OLEA, [C_17_H_19_O_6_]^−^			
0 D (319.1187)	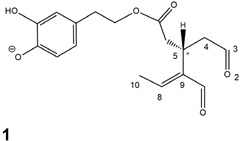	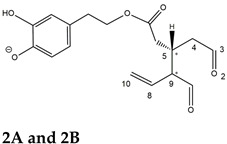	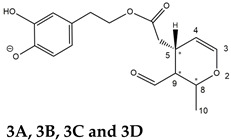
1 D (320.1250)	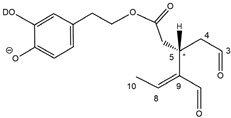	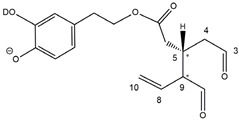	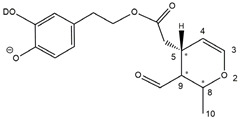
2 D (321.1313)	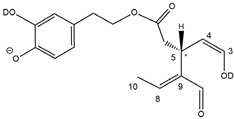	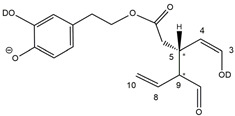	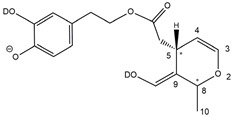
3 D (322.1375)	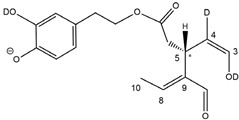	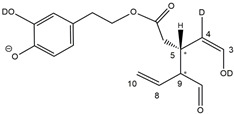	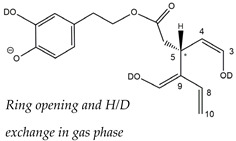
4 D (323.1438)		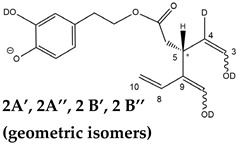	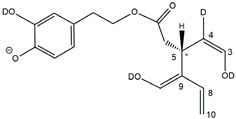

## Data Availability

The data presented in this study are available on request from the corresponding author.

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
