# Peer review of "Hydrogen/Deuterium Exchange Mass Spectrometry for Probing the Isomeric Forms of Oleocanthal and Oleacin in Extra Virgin Olive Oils"

_molecules, 2023, doi:10.3390/molecules28052066_

Round 1

Reviewer 1 Report

The manuscript entitled (Hydrogen/Deuterium exchange mass spectrometry for probing 2 the isomeric forms of oleocanthal and oleacin in extra virgin ol- 3 ive oils) can be accepted for publication in Molecules journal after revision.

 The method for characterization is well presented and well interpretation of data was done, but

1-      The conclusion part is missing from the manuscript

2-      The result and discussion both are too long with some sort of redundancy please, make them more specific and simpler.

3-      You have to state your novelty and advantages over the published methods

4-      It would be very interesting to apply your proposed method on real olive oil samples taken from the market

Author Response

Reviewer #1

The manuscript entitled (Hydrogen/Deuterium exchange mass spectrometry for probing the isomeric forms of oleocanthal and oleacin in extra virgin olive oils) can be accepted for publication in Molecules journal after revision.

Reply: we thank the Reviewer for his/her comments on our paper.

The method for characterization is well presented and well interpretation of data was done, but

1-      The conclusion part is missing from the manuscript

Reply: we followed the instructions to Authors provided by the journal, stating that the Conclusion section can be skipped if the discussion includes the main findings of a manuscript without being too long. Anyway, following the Reviewer’s suggestion, a Conclusion section, highlighting the most important results obtained during the study, has been added to the revised version of the paper.

2-      The result and discussion both are too long with some sort of redundancy please, make them more specific and simpler.

Reply: the Discussion section was revised to reduce the redundancy with the Results section. Further considerations were added to the Discussion section on the comparison between the outcomes of the present study and those reported in the literature.

3-      You have to state your novelty and advantages over the published methods                                                                                                                                          

Reply: we have emphasised the novelty of our study both in the revised Discussion section and in the Conclusion section, that was added to the revised version of the paper.

4-      It would be very ineresting to apply your proposed method on real olive oil samples taken from the market

Reply: we would like to point out that both the olive oil used for the present study and those for which data on oleacin and oleocanthal were reported in our previous papers were all real olive oils, obtained in different regions of Italy and produced for commercialization.

Reviewer 2 Report

This manuscript investigate the structural details of isomeric forms of oleocanthal and oleacin in extra virgin olive oils based on the hydrogen/Deuterium exchange mass spectrometry, It is helpful to understand the composition and further study its biological activity.

1.  A Q-Exactive quadrupole-Orbitrap mass spectrome-ter (Thermo Scientific, Waltham, MA, USA) was employed. Is it a Fourier-trans-form single and tandem mass spectrometry (RPLC-ESI-FTMS and FTMS/MS)?

2. Whether standard products can be obtained, if so the relevant spectrum of the standard should be supplemented and compared with them in EVOO sample to provide more evidence.

3. In figure 3A, m/z 319.11670-303.12070, The latter may be wrong.

4. The 2A and 2B are not the same compounds in figure 2 and figure 3, It is suggested to mark with different letters.

Author Response

Reviewer #2

This manuscript investigate the structural details of isomeric forms of oleocanthal and oleacin in extra virgin olive oils based on the hydrogen/Deuterium exchange mass spectrometry, It is helpful to understand the composition and further study its biological activity.

Reply: we thank the Reviewer for his/her comments on our paper.

1. A Q-Exactive quadrupole-Orbitrap mass spectrometer (Thermo Scientific, Waltham, MA, USA) was employed. Is it a Fourier-transform single and tandem mass spectrometry (RPLC-ESI-FTMS and FTMS/MS)?

Reply: yes. Like in all mass spectrometers including an Orbitrap mass analyzer, Fourier transform is adopted to process the “image current” generated by ions moving inside the Orbitrap in the case of the Q-Exactive spectrometer.

2. Whether standard products can be obtained, if so the relevant spectrum of the standard should be supplemented and compared with them in EVOO sample to provide more evidence.

Reply: we than the Reviewer for his/her suggestion. We were able to analyse a mixture of standards of oleocanthal and oleacin, recently made available commercially, using the same analytical approach described for the olive oil sample. As emphasized by Figure S5, added to the revised Supplementary Material, and as discussed in the revised paper, the profiles resulting from ion extraction for the two standard secoiridoids were almost the same as those observed for the olive oil extract, thus confirming that the observed features did not arise from eventual artifacts related to the extraction procedure.

3. In figure 3A, m/z 319.11670-303.12070, The latter may be wrong.

Reply: we thank the Reviewer for pointing out the typing error occurred when reporting the right end of the m/z interval. The error was corrected.

4. The 2A and 2B are not the same compounds in figure 2 and figure 3, It is suggested to mark with different letters.

Reply: in this case we used the same letters to emphasize the correspondence, in terms of structural features and, consequently, of retention order, between specific isoforms of oleocanthal and oleacin, thus we would prefer keeping that nomenclature in the paper and in the figures.

Reviewer 3 Report

This paper is a good contribution to the literature, but it is quite heavy for a reader not deep in mass spectrometry. The paragraph of the conclusions is missing and that of the discussions does not provide a real comparison with the literature. I would suggest to the authors join the results and discussion paragraphs. Furthermore, in line 57 there is a typing error (plant not plat), and in lines 670 and 678 the sign -, after the closing of the square bracket, it should be like superscript.

Author Response

Reviewer #3

This paper is a good contribution to the literature, but it is quite heavy for a reader not deep in mass spectrometry. The paragraph of the conclusions is missing and that of the discussions does not provide a real comparison with the literature. I would suggest to the authors join the results and discussion paragraphs. Furthermore, in line 57 there is a typing error (plant not plat), and in lines 670 and 678 the sign -, after the closing of the square bracket, it should be like superscript.

Reply: we thank the Reviewer for his/her comments on our paper. We realize that the description of results obtained from LC-MS analysis can be complex, especially for the H/D exchange experiments. We tried to explain the results as simply as possible. 

Following the Reviewer’s suggestion, a Conclusion section, highlighting the most important results obtained during the study, has been added to the revised version of the paper and a more detailed comparison with the literature has been made in the Discussion section. We would prefer keeping the two sections as distinct parts.

We thank the Reviewer for pointing out the typing errors, that were promptly corrected.

Round 2

Reviewer 1 Report

The review is accepted for publication in its present form